# CLASS-CONTEXT-AWARE PHANTOM UNCERTAINTY MODELING

## ABSTRACT

Uncertainty modeling is crucial in developing robust and reliable models since it enables decision-makers to access the trustworthiness of predictions and make informed choices based on the uncertainty associated with the prediction. A straightforward approach to endow models with the ability to estimate uncertainty involves modeling a probabilistic distribution of the input representations and approximating it by variational inference. However, this method inevitably leads to an issue where uncertainty is underestimated, resulting in overconfident predictions even when dealing with data that contains inherent noise or ambiguity. In response to this challenge, we introduce a novel approach called Class-Context-aware Phantom Uncertainty Modeling. To circumvent the problem of underestimating uncertainty associated with the input data, we shift the focus to infer the distribution of their respective phantoms, which are derived by leveraging class-contextual information. We mitigate the issue of uncertainty underestimation by demonstrating that the estimated uncertainty of the original input data is no less than that of the phantom. We showcase our method's superior robustness and generalization capabilities through experiments involving robust learning tasks such as noisy label learning and cross-domain generalization.

## 1 INTRODUCTION

Conventional deep neural networks are deterministic, primarily focused on producing a single point estimate of the prediction Krizhevsky et al. (2012); LeCun et al. (2015); He et al. (2016). While they have revolutionized various fields, deterministic networks are often prone to assigning overconfident predictions to out-of-distribution (OOD) inputs Guo et al. (2017); Hein et al. (2019) and incapable of assessing the inherent ambiguity in the input data (a.k.a. aleatoric uncertainty) Hüllermeier & Waegeman (2021). These limit their reliability in real-world scenarios and may result in disastrous consequences in safety-critical tasks, such as autonomous driving and medical diagnosis Kendall & Gal (2017).

For this reason, researchers have begun incorporating uncertainty modeling in deep networks Gal (2016). By producing a probabilistic distribution of representations Ghahramani (2015), deep networks are equipped with the capability to capture the aleatoric uncertainty of the input data, with a larger variance indicating higher uncertainty. This allows networks to adjust overconfident predictions when confronted with unfamiliar inputs Chang et al. (2020), consequently enhancing their generalization accuracy, especially when these models are exposed to OOD data Li et al. (2022).

Although modeling uncertainty by probabilistic representations is promising, a fundamental challenge emerges that there is no guarantee that the moments of the distribution can be accurately inferred, due to the absence of uncertainty ground truth for real data. In fact, empirical evidence suggests that the powerful fitting capability of deep networks can adversely affect the estimation of variance such that the uncertainty shrinks rapidly during training Zhang et al. (2021), making it deteriorate to deterministic representations. This is attributed to the fact that probabilistic distributions are approximated using variation inference, which tends to underestimate the posterior variance by memorizing fixed locations in the representation space, a phenomenon that has been extensively studied in the literature Wang & Titterington (2005); Blei et al. (2017); Zhao et al. (2019).

To alleviate the variance underestimation issue, we propose a novel approach that centers on creating and inferring a "phantom" entity in the representation space. Specifically, given an input, we

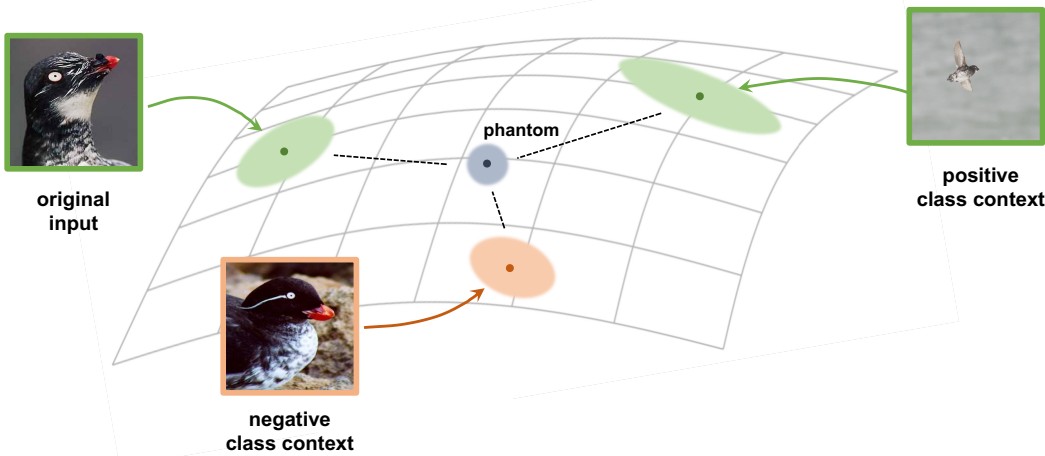

Figure 1: The diagram illustrates the concept of our class-context-aware phantom. When considering the original input, which is labeled as "Least Auklet Parakeet Auklet", two related class-contextual samples are identified. The "positive class context", also labeled as "Least Auklet Parakeet Auklet", exhibits the largest intra-class variation, while the "negative class context", labeled as "Parakeet Auklet", exhibits the smallest inter-class variation. All three of these components contribute to generating a probabilistic distribution within the latent space, and the phantom is subsequently constructed using the moment parameters derived from these distributions. In theory, the variance of the phantom exhibits a characteristic where it is lower than the variance of the original input (by Eq. 9).

first identify its class context samples that demonstrate the most significant within-class variability (referred to as the *positive class context*) and the least between-class variability (referred to as the *negative class context*). These class context samples provide strong discriminative information. It is also worth noting that these class context samples tend to yield higher levels of uncertainty. This is due to the fact that 1) positive class contexts are apart from the input in the latent space, necessitating a larger variance to encompass the range of variations and align with the representation distribution of the original input; and 2) negative class contexts are in close proximity to the input in the latent space, however, in order to prevent overconfident yet incorrect prediction, a larger variance is required to allow them to be resampled to a more distant location to facilitate separation. It is to be emphasized that both types of class context samples are found in the training data. Per their definition, they have a tendency to produce a larger variance estimate of the distribution in the representation space by enhancing awareness of the potential variability within each class, thus helping alleviate the uncertainty underestimation issue from this perspective.

Once the class-contextual samples are identified, the phantom of the original input is constructed by the combination of itself and its class-contextual samples, as illustrated in Figure 1. Specifically, the variance of the phantom distribution adheres to the structure of the Best Linear Unbiased Estimation (BLUE) Henderson (1975), while the mean is derived by blending the representations of the original input and its class context samples. Once the phantom is established, we proceed to employ the variation inference technique to infer the distribution of the phantoms. In our VI framework, the phantom distribution is adopted as the variational distribution $q$ to be learned to approximate the unknown true distribution $p$ by regularizing the KL divergence. Therefore, we avoid the variation inference of the original input representation, which could lead to variance underestimation. Moreover, according to the structure of the BLUE (Eq. 9), a key insight is that the variance of the phantom serves as a lower bound for the variance of the original input representation. As its name suggests, the phantom does not correspond to any samples in the input space. Consequently, this provides another perspective: our approach mitigates the uncertainty underestimation issue by shifting the VI optimization objective from the original input to its phantom.

Our main contributions are three-fold:

1. We leverage the informative merits of both positive and negative class context samples to construct a novel "phantom" in the representation space.

2. Through VI applied to the created phantom distribution, our approach establishes a lower bound for the estimated uncertainty of each input. This effectively mitigates the problem of uncertainty underestimation, which leads to overconfident predictions even for noisy or ambiguous input.

3. Extensive experiments on robust learning tasks demonstrate our method's strong generalization and robustness when handling label noise and out-of-domain data.

## 2 RELATED WORK

**Uncertainty Modeling.** Uncertainty can manifest in different forms, including *epistemic uncertainty* and *aleatoric uncertainty* Der Kiureghian & Ditlevsen (2009); Kendall & Gal (2017); Hüllermeier & Waegeman (2021). Epistemic uncertainty (a.k.a. model uncertainty) is incurred by a lack of knowledge, usually due to insufficient training data. To model epistemic uncertainty, it often relies on capturing the stochasticity of model parameters by Monte-Carlo sampling Blundell et al. (2015); Gal & Ghahramani (2016); Lakshminarayanan et al. (2017), which could be computationally expensive. On the other hand, our focus is more on aleatoric uncertainty (a.k.a. data uncertainty), which mainly comes from intrinsic noise or ambiguity in the data caused by various real-world factors, such as label noise, OOD outliers and unknown domain shifts.

In the deep learning era, capturing aleatoric uncertainty usually relies on the power of neural networks to parameterize the moments of representation distributions. For example, Kendall & Gal (2017) predicted the mean and the variance with the negative log-likelihood loss function. Oh et al. (2018) proposed to estimate the moments by distribution matching. Shi & Jain (2019) further introduced the Mutual Likelihood Score to facilitate the matching. Chang et al. (2020) proposed to learn the mean and variance simultaneously by variational information bottleneck and showcased the benefit of handling noisy data.

Our approach is significantly different from previous uncertainty modeling methods. This is due to the fact that we do not treat each input independently to determine its uncertainty, as a common practice of most previous methods. On the contrary, we introduce a novel phantom uncertainty modeling approach that considers the class variability information provided by the class-contextual samples. This brings the notion of uncertainty closer to a contrasting concept rather than an isolated one.

**Variational Inference.** VI has long-standingly played an important role in the realm of probabilistic modeling and uncertainty estimation Jordan et al. (1999); Bishop & Nasrabadi (2006); Blei et al. (2017). The core concept of VI revolves around approximating challenging-to-compute distributions through optimization techniques. While it has primarily been associated with generative models, such as in the case of Kingma and Welling's Auto-Encoding Variational Bayes Kingma & Welling (2013), VI has also left its imprint on discriminative models. For example, VIB Alemi et al. (2017) cast the problem of information bottleneck as a VI problem by optimizing the encoding distribution to minimize the mutual information between the input and the encoded representation.

Nevertheless, due to its optimization objective, VI is susceptible to issues such as posterior collapse or the underestimation of variance. Notably, we differ from most previous VI methods, such that we perform inference on the variational distribution pertaining to the constructed phantoms rather than the actual data points. The variance linked to these phantoms establishes a minimum threshold for estimating the actual variance of interest.

## 3 METHODOLOGY

In this section, we demonstrate our method of Class-Context-aware Phantom Uncertainty Modeling in detail. We first introduce the necessary preliminaries for the problem formulation in Sec. 3.1. Then we propose our idea in Sec. 3.2. Finally, we derive the loss function to train our method in Sec. 3.3.

## 3.1 PRELIMINARIES: UNCERTAINTY MODELING AND ITS UNDERESTIMATION

We consider the general task of multi-class classification with $K$ labels. For training, a batch of training samples $\mathcal{B}_n = \{(\boldsymbol{x}_i, y_i)\}_{i=1}^n$ is given, where each input-output pair $(\boldsymbol{x}_i, y_i)$ is drawn from a probability distribution over the sample space $\mathcal{X} \times \mathcal{Y}$. Here, $\mathcal{X} \subset \mathbb{R}^D$ denotes the input space of dimension $D$ and $\mathcal{Y} = \{1, ..., K\}$ denotes the set of output labels. A deep neural network is to be trained, which comprises an $L$ layers embedding function $f : \mathcal{X} \to \mathbb{R}^d$ that maps an input $\boldsymbol{x}$ to its representation $\boldsymbol{z} \in \mathbb{R}^d$, and a classifier $g : f(\mathcal{X}) \to \mathcal{Y}$ that maps the representation to the label prediction. Formally, the prediction is produced by using the softmax function:

$$\boldsymbol{z} = f(\boldsymbol{x}), \quad q(y = k|\boldsymbol{z}) = \frac{\exp\{g_k(\boldsymbol{z})\}}{\sum_c \exp\{g_c(\boldsymbol{z})\}}. \tag{1}$$

However, in practical scenarios, random errors can occur during the data acquisition process, resulting in intrinsic data noise. As a result, the relationship between $\mathcal{X}$ and $\mathcal{Y}$ tends to exhibit a probabilistic nature rather than a deterministic one. Therefore, instead of providing a deterministic representation, it is more desirable to produce a probabilistic one that accounts for the uncertainty Nix & Weigend (1994); Bishop (1994). To represent this probabilistic distribution and capture the uncertainty, a common practice is to employ an isotropic Gaussian distribution Kendall & Gal (2017). Specifically, once an input $\boldsymbol{x}$ is processed through the penultimate layer of $f$ to obtain feature maps $f^{L-1}(\boldsymbol{x})$, a two-branch architecture is appended on top of this penultimate layer to predict a mean vector $\boldsymbol{\mu} \in \mathbb{R}^d$ and a variance vector $\boldsymbol{\sigma}^2 \in \mathbb{R}^d$ (which is always positive):

$$\boldsymbol{\mu} = f_{\theta_{\boldsymbol{\mu}}}(f^{L-1}(\boldsymbol{x})), \quad \boldsymbol{\sigma}^2 = f_{\theta_{\boldsymbol{\sigma}^2}}(f^{L-1}(\boldsymbol{x})), \tag{2}$$

where $\theta_{\boldsymbol{\mu}}$ and $\theta_{\boldsymbol{\sigma}^2}$ refer to the model parameters w.r.t. output $\boldsymbol{\mu}$ and $\boldsymbol{\sigma}^2$ respectively. In this manner, the representation of each input follows a multivariate Gaussian distribution:

$$q(\boldsymbol{z}|\boldsymbol{x}) = \mathcal{N}(\boldsymbol{z}; \boldsymbol{\mu}, \boldsymbol{\sigma}^2 \boldsymbol{I}). \tag{3}$$

By training the neural network to estimate the parameters $\boldsymbol{\mu}$ and $\boldsymbol{\sigma}^2$ based on the input data $\boldsymbol{x}$, the model is equipped with the capability of quantifying the uncertainty of its prediction. Inspired by variational inference methods Blei et al. (2017); Alemi et al. (2017), both $\boldsymbol{\mu}$ and $\boldsymbol{\sigma}^2$ can be learned by maximizing the evidence lower bound (ELBO) of the predictive distribution, as follows:

$$\log p(y|\boldsymbol{x}) \geq \int q(\boldsymbol{z}|\boldsymbol{x}) \log q(y|\boldsymbol{z}) d\boldsymbol{z} - \mathrm{KL}\left(q(\boldsymbol{z}|\boldsymbol{x})||p(\boldsymbol{z}|\boldsymbol{x})\right)$$
$$= \int \mathcal{N}(\boldsymbol{z}; \boldsymbol{\mu}, \boldsymbol{\sigma}^2 \boldsymbol{I}) \log \frac{\exp\left\{\boldsymbol{z}^T \boldsymbol{w}_y + b_y\right\}}{\sum_c \exp\left\{\boldsymbol{z}^T \boldsymbol{w}_c + b_c\right\}} d\boldsymbol{z} - \mathrm{KL}(\mathcal{N}(\boldsymbol{z}; \boldsymbol{\mu}, \boldsymbol{\sigma}^2 \boldsymbol{I})||\mathcal{N}(\boldsymbol{z}; \boldsymbol{0}, \boldsymbol{I})), \tag{4}$$

where the classifier $g$ is represented by a linear layer parameterized by $\boldsymbol{w}$ and $b$, with each subscript indicating its association with a specific class, and $p(\boldsymbol{z}|\boldsymbol{x})$ is a conditional prior characterized by a standard Gaussian.

However, the variance $\boldsymbol{\sigma}^2$ is known to be underestimated, primarily due to the mean-field assumption that the posterior distribution can be factorized (as expressed by the vectorization of $\boldsymbol{\sigma}^2$). While this simplification makes inference computationally tractable, the KL divergence term in Eq. 4 penalizes regions in latent space where $q(\boldsymbol{z}|\boldsymbol{x})$ has high density but remains indifferent to regions where $q(\boldsymbol{z}|\boldsymbol{x})$ exhibits low density. Consequently, variational inference has the potential to yield a posterior distribution with reduced variance. These class-contextual samples inevitably deteriorate the convergence of model training, thus avoiding making overconfident predictions.

## 3.2 CLASS-CONTEXT-AWARE PHANTOM UNCERTAINTY MODELING

To alleviate the aforementioned uncertainty underestimation issue, instead of relaxing the mean-field assumption which may lead to expensive computation, we propose to model the uncertainty associated with the class-context-aware phantoms. The graphical model is depicted in Fig. 2. Here, we provide a formal definition of the class-contextual samples and phantoms. For each input $\boldsymbol{x}$ with

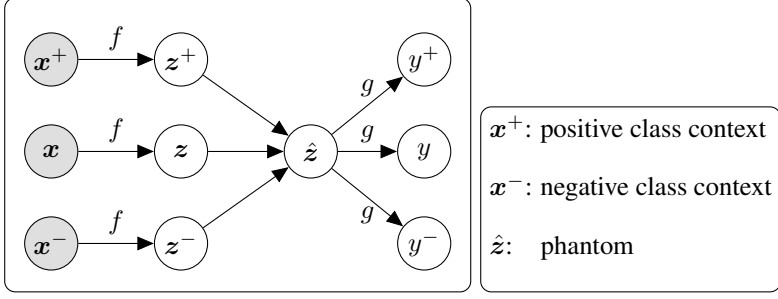

Figure 2: The graphical model of our method.

corresponding class label $y$, the class-contextual samples $\boldsymbol{x}^+$ and $\boldsymbol{x}^-$ are derived as follows:

$$
\begin{aligned}
\boldsymbol{x}^+ &= \underset{\boldsymbol{x}_j \in \mathcal{B}_n,\ \text{s.t.}\ y_j=y}{\operatorname{argmax}} ||f_{\theta_{\boldsymbol{\mu}}}(f^{L-1}(\boldsymbol{x}_j)) - \boldsymbol{\mu}||, \\
\boldsymbol{x}^- &= \underset{\boldsymbol{x}_k \in \mathcal{B}_n\ \text{s.t.}\ y_k \neq y}{\operatorname{argmin}} ||f_{\theta_{\boldsymbol{\mu}}}(f^{L-1}(\boldsymbol{x}_k)) - \boldsymbol{\mu}||,
\end{aligned}
\tag{5}
$$

where $\boldsymbol{\mu}$ is the mean of the representation of $\boldsymbol{x}$ obtained by Eq. 2. Now, we delve into why these class-contextual samples provide useful discriminative information: 1) the positive class context $\boldsymbol{x}^+$, originating from the same class but visually distinct from $\boldsymbol{x}$, serves to capture the variations within the same class; 2) the negative class context $\boldsymbol{x}^-$, visually similar to $\boldsymbol{x}$ but from a different class, helps discern the boundaries between classes.

Furthermore, these class contextual samples exhibit a tendency to produce greater uncertainties. This occurs since the positive class context $\boldsymbol{x}^+$ requires a larger variance to encompass the range of variations and overlap with the distribution of the input representation $\mathcal{N}(\boldsymbol{z}; \boldsymbol{\mu}, \boldsymbol{\sigma}^2 \boldsymbol{I})$. Similarly, the negative class context $\boldsymbol{x}^-$ also demands a greater variance to enable its resampling to more distant locations for better separation.

After the class-contextual samples are obtained, we can denote the probabilistic representations of $\boldsymbol{x}, \boldsymbol{x}^+$, and $\boldsymbol{x}^-$ as $\mathcal{N}(\boldsymbol{\mu}, \boldsymbol{\sigma}^2)$, $\mathcal{N}(\boldsymbol{\mu}^+, (\boldsymbol{\sigma}^+)^2)$ and $\mathcal{N}(\boldsymbol{\mu}^-, (\boldsymbol{\sigma}^-)^2)$ respectively. Next, we define the phantom $\hat{\boldsymbol{z}}$ from these distributions. To realize this, we maximize the ELBO of the triplet predictive distribution as follows (Proof of Eq. 6 can be found in Appendix A):

$$
\begin{aligned}
&\log p(y, y^+, y^- | \boldsymbol{x}, \boldsymbol{x}^+, \boldsymbol{x}^-) \\
&\geq \int q(\hat{\boldsymbol{z}}|\boldsymbol{x}, \boldsymbol{x}^+, \boldsymbol{x}^-) \log p(y, y^+, y^-|\hat{\boldsymbol{z}}) d\hat{\boldsymbol{z}} + \int q(\hat{\boldsymbol{z}}|\boldsymbol{x}, \boldsymbol{x}^+, \boldsymbol{x}^-) \log \frac{p(\hat{\boldsymbol{z}}|\boldsymbol{x}, \boldsymbol{x}^+, \boldsymbol{x}^-)}{q(\hat{\boldsymbol{z}}|\boldsymbol{x}, \boldsymbol{x}^+, \boldsymbol{x}^-)} d\hat{\boldsymbol{z}}.
\end{aligned}
\tag{6}
$$

We call $\hat{\boldsymbol{z}}$ as the phantom. By assuming that the latent variables $\boldsymbol{z}, \boldsymbol{z}^+, \boldsymbol{z}^-$ are conditionally independent of each other given the input variables $\boldsymbol{x}, \boldsymbol{x}^+, \boldsymbol{x}^-$, the distribution of phantom can be factorized as:

$$
\begin{aligned}
q(\hat{\boldsymbol{z}}|\boldsymbol{x}, \boldsymbol{x}^+, \boldsymbol{x}^-) &\propto q(\hat{\boldsymbol{z}}|\boldsymbol{z}, \boldsymbol{z}^+, \boldsymbol{z}^-) \cdot q(\boldsymbol{z}, \boldsymbol{z}^+, \boldsymbol{z}^-|\boldsymbol{x}, \boldsymbol{x}^+, \boldsymbol{x}^-) \\
&= \delta_{\hat{\boldsymbol{z}}-h(\boldsymbol{z}, \boldsymbol{z}^+, \boldsymbol{z}^-)} \cdot q(\boldsymbol{z}|\boldsymbol{x}) \cdot q(\boldsymbol{z}^+|\boldsymbol{x}^+) \cdot q(\boldsymbol{z}^-|\boldsymbol{x}^-).
\end{aligned}
\tag{7}
$$

Here $\delta$ denotes the Dirac delta function and $h(\boldsymbol{z}, \boldsymbol{z}^+, \boldsymbol{z}^-)$ is blending function of $\boldsymbol{z}, \boldsymbol{z}^+, \boldsymbol{z}^-$. $q(\hat{\boldsymbol{z}}|\boldsymbol{z}, \boldsymbol{z}^+, \boldsymbol{z}^-)$ can be written as a Dirac delta function since $\hat{\boldsymbol{z}}$ is a deterministic function of $\boldsymbol{z}, \boldsymbol{z}^+$ and $\boldsymbol{z}^-$. To calculate the product $q(\boldsymbol{z}|\boldsymbol{x}) \cdot q(\boldsymbol{z}^+|\boldsymbol{x}^+) \cdot q(\boldsymbol{z}^-|\boldsymbol{x}^-)$, we adopt the following lemma.

**Lemma 3.1 (Product of Gaussian distributions)** *Denote $p_1(x), p_2(x)$ and $p_3(x)$ are Gaussian density functions with means and standard deviations as $\mu_i$ and $\sigma_i^2, i = 1, 2, 3$. Then the product of the Gaussian distributions are proportional to a new Gaussian distribution $\mathcal{N}(\hat{\mu}, \hat{\sigma}^2)$ as follows:*

$$
\hat{\mu} = \hat{\sigma}^2 \left( \frac{\mu_1}{\sigma_1^2} + \frac{\mu_2}{\sigma_2^2} + \frac{\mu_3}{\sigma_3^2} \right), \quad \frac{1}{\hat{\sigma}^2} = \frac{1}{\sigma_1^2} + \frac{1}{\sigma_2^2} + \frac{1}{\sigma_3^2}.
\tag{8}
$$

The proof can be found in Bromiley (2003). Note that $q(\boldsymbol{z}|\boldsymbol{x})$, $q(\boldsymbol{z}^+|\boldsymbol{x}^+)$ and $q(\boldsymbol{z}^-|\boldsymbol{x}^-)$ are assumed to be Gaussian distributions $\mathcal{N}(\boldsymbol{\mu}, \boldsymbol{\sigma}^2\boldsymbol{I})$, $\mathcal{N}(\boldsymbol{\mu}^+, (\boldsymbol{\sigma}^+)^2\boldsymbol{I})$ and $\mathcal{N}(\boldsymbol{\mu}^-, (\boldsymbol{\sigma}^-)^2\boldsymbol{I})$ respectively. By Lemma 3.1, the phantom still follows a (scaled) Gaussian $q(\hat{\boldsymbol{z}}|\boldsymbol{x}, \boldsymbol{x}^+, \boldsymbol{x}^-) \propto \mathcal{N}(\hat{\boldsymbol{\mu}}, \hat{\boldsymbol{\sigma}}^2)$, with the optimal parameters deduced by:

$$\begin{aligned}
\hat{\boldsymbol{\mu}} &= \boldsymbol{w} \odot \boldsymbol{\mu} + \boldsymbol{w}^+ \odot \boldsymbol{\mu}^+ + \boldsymbol{w}^- \odot \boldsymbol{\mu}^-, \\
\hat{\boldsymbol{\sigma}}^2 &= \boldsymbol{\sigma}^{-2} + (\boldsymbol{\sigma}^+)^{-2} + (\boldsymbol{\sigma}^-)^{-2},
\end{aligned} \tag{9}$$

where the weights $\boldsymbol{w}$, $\boldsymbol{w}^+$ and $\boldsymbol{w}^-$ are uncertainty-aware as follows:

$$\begin{aligned}
\boldsymbol{w} &= \boldsymbol{\sigma}^{-2} \oslash (\boldsymbol{\sigma}^{-2} + (\boldsymbol{\sigma}^+)^{-2} + (\boldsymbol{\sigma}^-)^{-2}), \\
\boldsymbol{w}^+ &= (\boldsymbol{\sigma}^+)^{-2} \oslash (\boldsymbol{\sigma}^{-2} + (\boldsymbol{\sigma}^+)^{-2} + (\boldsymbol{\sigma}^-)^{-2}), \\
\boldsymbol{w}^- &= (\boldsymbol{\sigma}^-)^{-2} \oslash (\boldsymbol{\sigma}^{-2} + (\boldsymbol{\sigma}^+)^{-2} + (\boldsymbol{\sigma}^-)^{-2}).
\end{aligned} \tag{10}$$

Note that $\odot$ and $\oslash$ represent the element-wise Hadamard multiplication and division, respectively.

The term "phantom" is used because it describes an artificial distribution in the latent space that does not correspond to any real data in the input space. The variance $\hat{\boldsymbol{\sigma}}^2$ associated with the phantom (in Eq. 9) follows a similar structure to the Best Linear Unbiased Estimator (BLUE), where it is lower bounded by the uncertainty of representations of the input $\boldsymbol{x}$ and its class contexts. By shifting the focus of uncertainty estimation from the input $\boldsymbol{x}$ alone (by solving Eq. 4) to this phantom distribution, we can mitigate the issue of variance underestimation regarding the uncertainty of $\boldsymbol{x}$.

For the second term in Eq. 6, it can be divided into the following (Proof of Eq. 11 can be found in Appendix B):

$$\begin{aligned}
&\int q(\hat{\boldsymbol{z}}|\boldsymbol{x}, \boldsymbol{x}^+, \boldsymbol{x}^-) \log \frac{p(\hat{\boldsymbol{z}}|\boldsymbol{x}, \boldsymbol{x}^+, \boldsymbol{x}^-)}{q(\hat{\boldsymbol{z}}|\boldsymbol{x}, \boldsymbol{x}^+, \boldsymbol{x}^-)} d\hat{\boldsymbol{z}} \\
&= \int q(\boldsymbol{z}, \boldsymbol{z}^+, \boldsymbol{z}^-|\boldsymbol{x}, \boldsymbol{x}^+, \boldsymbol{x}^-) \log \frac{p(\boldsymbol{z}, \boldsymbol{z}^+, \boldsymbol{z}^-|\boldsymbol{x}, \boldsymbol{x}^+, \boldsymbol{x}^-)}{p(\boldsymbol{z}|\boldsymbol{x})p(\boldsymbol{z}^+|\boldsymbol{x}^+)p(\boldsymbol{z}^-|\boldsymbol{x}^-)} d\boldsymbol{z} d\boldsymbol{z}^+ d\boldsymbol{z}^- - \text{KL}[q(\hat{\boldsymbol{z}}|\boldsymbol{x}, \boldsymbol{x}^+, \boldsymbol{x}^-)||p(\boldsymbol{z}|\boldsymbol{x})] \\
&\quad - \text{KL}[q(\hat{\boldsymbol{z}}|\boldsymbol{x}, \boldsymbol{x}^+, \boldsymbol{x}^-)||p(\boldsymbol{z}^+|\boldsymbol{x}^+)] - \text{KL}[q(\hat{\boldsymbol{z}}|\boldsymbol{x}, \boldsymbol{x}^+, \boldsymbol{x}^-)||p(\boldsymbol{z}^-|\boldsymbol{x}^-)].
\end{aligned} \tag{11}$$

Here, the first term in Eq. 11 represents the mutual influences among the latent variables. Unfortunately, this integral poses a computational challenge due to the absence of an analytical expression for $p$. However, we introduce a valid assumption, grounded in the definitions of $\boldsymbol{z}$, $\boldsymbol{z}^+$, and $\boldsymbol{z}^-$, which aims to bring positive representations closer together and push negative representations farther apart. Consequently, we can approximate it in a contrastive manner:

$$\int q(\boldsymbol{z}, \boldsymbol{z}^+, \boldsymbol{z}^-|\boldsymbol{x}, \boldsymbol{x}^+, \boldsymbol{x}^-) \log \frac{p(\boldsymbol{z}, \boldsymbol{z}^+, \boldsymbol{z}^-|\boldsymbol{x}, \boldsymbol{x}^+, \boldsymbol{x}^-)}{p(\boldsymbol{z}|\boldsymbol{x})p(\boldsymbol{z}^+|\boldsymbol{x}^+)p(\boldsymbol{z}^-|\boldsymbol{x}^-)} d\boldsymbol{z} d\boldsymbol{z}^+ d\boldsymbol{z}^- \approx \log \frac{\exp\{\boldsymbol{z} \cdot \boldsymbol{z}^+/\tau\}}{\exp\{\boldsymbol{z} \cdot \boldsymbol{z}^+/\tau + \boldsymbol{z} \cdot \boldsymbol{z}^-/\tau\}}. \tag{12}$$

For the K-L divergence terms in Eq. 11, following Alemi et al. (2017), $p(\boldsymbol{z}|\boldsymbol{x})$, $p(\boldsymbol{z}^+|\boldsymbol{x}^+)$ and $p(\boldsymbol{z}^-|\boldsymbol{x}^-)$ are assumed to follow a standard Gaussian distribution $\mathcal{N}(\boldsymbol{0}, \boldsymbol{I})$.

### 3.3 LOSS FUNCTION

To derive the overall objective function to train our method, we minimize the negative of our triplet ELBO in Eq. 6. For the first term, to make the gradient tractable, the re-parametrization trick is adopted Kingma & Welling (2013). We use $\hat{\boldsymbol{z}}_{\boldsymbol{\epsilon}} = \hat{\boldsymbol{\mu}} + \boldsymbol{\epsilon} \odot \hat{\boldsymbol{\sigma}}^2$ as the sample of posterior distribution $\mathcal{N}(\hat{\boldsymbol{\mu}}, \hat{\boldsymbol{\sigma}}^2)$, where $\boldsymbol{\epsilon}$ is a standard Gaussian noise. Then the negative of the first integral term in Eq. 6 can expressed as the cross entropy loss:

$$\begin{aligned}
\mathcal{L}_{\text{CE}}(\boldsymbol{x}) &= -\int q(\hat{\boldsymbol{z}}|\boldsymbol{x}, \boldsymbol{x}^+, \boldsymbol{x}^-) \log p(y, y^+, y^-|\hat{\boldsymbol{z}}) d\hat{\boldsymbol{z}} \\
&\simeq -\log \frac{\exp\{\hat{\boldsymbol{z}}_{\boldsymbol{\epsilon}}^T \boldsymbol{w}_y + b_y\}}{\sum_c \exp\{\hat{\boldsymbol{z}}_{\boldsymbol{\epsilon}}^T \boldsymbol{w}_c + b_c\}} - \log \frac{\exp\{\hat{\boldsymbol{z}}_{\boldsymbol{\epsilon}}^T \boldsymbol{w}_y^+ + b_y^+\}}{\sum_c \exp\{\hat{\boldsymbol{z}}_{\boldsymbol{\epsilon}}^T \boldsymbol{w}_c^+ + b_c^+\}} - \log \frac{\exp\{\hat{\boldsymbol{z}}_{\boldsymbol{\epsilon}}^T \boldsymbol{w}_y^- + b_y^-\}}{\sum_c \exp\{\hat{\boldsymbol{z}}_{\boldsymbol{\epsilon}}^T \boldsymbol{w}_c^- + b_c^-\}},
\end{aligned} \tag{13}$$

where $\boldsymbol{w}^+, b^+$ and $\boldsymbol{w}^-, b^-$ are the linear classifier parameters of the positive and negative context classes, respectively. This means our learned class-context-aware phantom $\hat{\boldsymbol{z}}$ should be discriminative enough to recognize $y$, $y^+$, and $y^-$ classes individually and jointly.

For the second term in Eq. 12, following Khosla et al. (2020), its negative can be approximated by the triplet margin loss (Proof can be found in Appendix C):

$$\mathcal{L}_{\text{TM}}(\boldsymbol{x}) = \max\left(\|\boldsymbol{\mu} - \boldsymbol{\mu}^+\|^2 - \|\boldsymbol{\mu} - \boldsymbol{\mu}^-\|^2 + 2\tau, 0\right), \tag{14}$$

where $2\tau$ is the margin parameter. The triplet loss is to enforce the compactness between samples with the same label and the separation between samples with different labels. Last but not least, the KL divergence terms in Eq. 11 have a tractable solution:

$$\mathcal{L}_{\text{KL}}(\boldsymbol{x}) = \text{KL}(\mathcal{N}(\hat{\boldsymbol{\mu}}, \hat{\boldsymbol{\sigma}}^2)\|\mathcal{N}(\boldsymbol{0}, \boldsymbol{I})) = -\frac{1}{2}(1 + \log\hat{\boldsymbol{\sigma}}^2 - \hat{\boldsymbol{\mu}}^2 - \hat{\boldsymbol{\sigma}}^2) \tag{15}$$

To sum up, the entire loss function of our method is as follows:

$$\mathcal{L} = \frac{1}{n}\left(\mathcal{L}_{\text{CE}}(\boldsymbol{x}) + \lambda_{\text{TM}}\mathcal{L}_{\text{TM}}(\boldsymbol{x}) + \lambda_{\text{KL}}\mathcal{L}_{\text{KL}}(\boldsymbol{x})\right). \tag{16}$$

The overall training algorithm of our proposed method is summarized in Algorithm. 1. As for testing, we do not need to leverage the class contextual samples or phantoms.

---

**Algorithm 1** Class-Context-aware Phantom Uncertainty Modeling

---

**Require:** A training batch $\mathcal{B}_n = \{(\boldsymbol{x}_i, y_i)\}_{i=1}^n$.
**Ensure:** A well-trained network with the ability to classify and estimate uncertainty.
 1: **for** each epoch **do**
 2:     Obtain class-context-aware class contexts $\boldsymbol{x}^+$ and $\boldsymbol{x}^-$ via Eq. 5;
 3:     Feedforward $\boldsymbol{x}$, $\boldsymbol{x}^+$ and $\boldsymbol{x}^-$ to estimate the probabilistic representations $\boldsymbol{z}$, $\boldsymbol{z}^+$ and $\boldsymbol{z}^-$;
 4:     Construct the phantom variable $\hat{\boldsymbol{z}}$ via Eq. 9;
 5:     Optimize the loss function in Equation 16 until convergence.
 6: **end for**

---

## 4 Experiments

By modeling the uncertainty of the class-context-aware phantoms, our method strives to prevent overconfident predictions, thereby enhancing the robustness when confronted with noisy or ambiguous data. To verify the effectiveness of our method, we conduct experiments on two common robust learning tasks, including noisy label learning (Sec. 4.1) and cross-domain generalization (Sec. 4.2).

All the experiments are run on a single NVIDIA GeForce 3090 GPU with 24GB memory. Inspired by the self-paced learning theory Kumar et al. (2010), we perform a "from-easy-to-hard" manner to mine the class contextual samples $\boldsymbol{x}^+$ and $\boldsymbol{x}^-$. Specifically, we initialize the class contextual samples as the randomly shuffled results of the mini-batch, and then the percentage of the class contextual samples to be exploited keeps increasing during training. For balancing parameters in our loss (Eq. 16), $\lambda_{\text{TM}}$ and $\lambda_{\text{KL}}$ are tuned to 3e-3 and 1e-4. The margin parameter for triplet margin loss is set to 1 by default. We report our results based on averaging over three random runs.

### 4.1 Noisy Label Learning

Learning from noisy labels presents a scenario where the data inherently contains noise or ambiguity. In such situations, the need for uncertainty modeling becomes essential to prevent the generation of overconfident yet incorrect predictions. We consider a real-world task, Facial Expression Recognition (FER), to verify the noisy label learning performance. In this task, a significant proportion of the face images exhibit inherent ambiguity due to annotator subjectivity and uncertainty introduced by face images in the wild. These factors make it particularly challenging to distinguish between various facial expressions accurately.

| Method | 0% Noise | 10% Noise | 20% Noise | 30% Noise |
|---|---|---|---|---|
| SCN Wang et al. (2020) | 87.35 | 81.92 | 80.02 | 77.46 |
| DUL Chang et al. (2020) | 88.04 | 85.08 | 81.95 | 78.90 |
| DMUE She et al. (2021) | 88.76 | 83.19 | 81.02 | 79.41 |
| RUL Zhang et al. (2021) | 88.98 | 86.22 | 84.34 | 82.06 |
| EAC Zhang et al. (2022b) | 89.99 | 88.02 | 86.05 | 84.42 |
| Ours | **90.03** | **88.69** | **86.38** | **85.32** |

Table 1: Testing accuracies on noisy facial expression recognition dataset RAF-DB.

| Method | 0% Reject | 10% Reject | 20% Reject | 30% Reject |
|---|---|---|---|---|
| SCN Wang et al. (2020) | 87.35 | 86.85 | 86.63 | 87.28 |
| DUL Chang et al. (2020) | 88.04 | 90.11 | 92.58 | 94.50 |
| RUL Zhang et al. (2021) | 88.33 | 91.80 | 94.54 | 96.32 |
| Ours | **90.03** | **92.58** | **94.74** | **96.46** |

Table 2: Testing accuracies on RAF-DB by rejecting different ratios of estimated uncertainty.

**Datasets**: We conducted experiments on one of the most widely utilized FER datasets, Real-world Affective Faces Database (RAF-DB Li et al. (2017b), which contains 29672 facial images annotated with basic and compound expressions. In our experiments, we adopt seven basic expressions (i.e., neutral, happiness, surprise, sadness, anger, disgust, and fear), including 12271 training images and 3068 test images. To introduce additional complexity and simulate real-world scenarios, we follow Chang et al. (2020); Wang et al. (2020); Zhang et al. (2021) to deliberately inject artificial label noise into the original training data by randomly choosing 10%, 20%, and 30% of samples and then flipping their labels to other random categories.

**Settings**: Following Zhang et al. (2021), all images in RAF-DB are resized to 224×224, and the ResNet-18 He et al. (2016) pretrained on Ms-Celeb-1M Guo et al. (2016) is used as the backbone network. To train our method, the training and testing batch size is 128, the total number of training epochs is 60, and the learning rate is initialized as 0.0004 and 0.004 for the backbone layers $f$ and the linear classifier layer $g$, respectively. Besides, an Adam optimizer Kingma & Ba (2014) with a weight decay of 0.0001 is used.

**Evaluation on Accuracy**: In Table 1, several state-of-the-art noise-tolerant facial expression recognition methods are compared. The same backbone of ResNet-18 and the pre-trained model are utilized for a fair comparison. Our method consistently outperforms the other methods under all circumstances by enhancing recognition accuracy under different ratios of noisy labels (0% Noise refers to using the original training data). Specifically, DUL Chang et al. (2020) is a baseline of our method that variationally optimizes the probabilistic representations for each input data independently by a cross-entropy loss and KL-divergence regularization. The significant improvement demonstrates the advantage of our method.

**Evaluation on Accuracy-Rejection Tradeoff**: To further verify that our method can predict meaningful uncertainty values and utilize them to identify the most uncertain data, we conduct experiments using the *accuracy versus rejection rate* metric Zhang et al. (2021): according to the uncertainty predictions, the images with top 10%, 20%, and 30% uncertain values are directly rejected, and the test accuracy is calculated on the remaining images. Therefore, the better a method models the uncertainty, the better testing accuracy can be achieved consistently when increasing the rejection ratio. The results are shown in Table 2. The results of the comparison methods are reported in Zhang et al. (2021). Our method again consistently outperforms all comparison methods under different rejection rates. This effectively demonstrates our method can learn more meaningful uncertainty values that are closely and consistently related to the final recognition confidence.

Table 3: Experiment results on the cross-domain image classification dataset PACS and Office-Home. (leave-one-domain-out)

| Method | PACS | | | | | Office-Home | | | | |
|---|---|---|---|---|---|---|---|---|---|---|
| | *Art* | *Cartoon* | *Photo* | *Sketch* | *Ave(%)* | *Art* | *Clipart* | *Product* | *Real* | *Ave(%)* |
| | ResNet-18 | | | | | ResNet-18 | | | | |
| Vanilla He et al. (2016) | 77.0 | 75.9 | 96.0 | 69.2 | 79.5 | 58.9 | 49.4 | 74.3 | 76.2 | 64.7 |
| Mixup Zhang et al. (2018) | 76.8 | 74.9 | 95.8 | 66.6 | 78.5 | 58.2 | 49.3 | 74.7 | 76.1 | 64.6 |
| L2A-OT Zhou et al. (2020) | 83.3 | 78.2 | 96.2 | 73.6 | 82.8 | 60.6 | 50.1 | 74.8 | 77.0 | 65.6 |
| SagNet Nam et al. (2021) | 83.6 | 77.7 | 95.5 | 76.3 | 83.3 | 60.2 | 45.4 | 70.4 | 73.4 | 62.3 |
| pAdaIN Nuriel et al. (2021) | 81.7 | 76.9 | 96.3 | 75.1 | 82.5 | — | — | — | — | — |
| MixStyle Zhou et al. (2021) | 82.3 | 79.0 | 96.3 | 73.8 | 82.8 | 58.7 | 53.4 | 74.2 | 75.9 | 65.5 |
| EFDM Zhang et al. (2022a) | 83.9 | 79.4 | 96.8 | 75.0 | 83.9 | — | — | — | — | — |
| DSU Li et al. (2022) | 83.6 | 79.6 | 95.8 | 77.6 | 84.1 | 60.2 | 54.8 | 74.1 | 75.1 | 66.1 |
| $I^2$-ADRMeng et al. (2022) | 82.9 | 80.8 | 95.0 | 83.5 | 85.6 | 66.4 | 53.3 | 74.9 | 75.3 | 67.5 |
| DCG Lv et al. (2023) | 86.0 | 80.8 | 96.4 | 82.1 | 86.3 | 60.7 | 55.5 | 75.3 | 76.8 | 67.1 |
| DomainDrop Guo et al. (2023) | 84.5 | 80.5 | 96.8 | 84.8 | 86.7 | 59.6 | 55.6 | 74.5 | 76.6 | 66.6 |
| **Ours** | 85.9 | 80.5 | 96.3 | 84.9 | **86.9** | 64.3 | 54.9 | 75.4 | 76.8 | **67.9** |
| | ResNet-50 | | | | | ResNet-50 | | | | |
| Vanilla He et al. (2016) | 84.4 | 77.1 | 97.6 | 70.8 | 82.5 | 61.3 | 52.4 | 75.8 | 76.6 | 66.5 |
| pAdaIN Nuriel et al. (2021) | 85.8 | 81.1 | 97.2 | 77.4 | 85.4 | — | — | — | — | — |
| MixStyle Zhou et al. (2021) | 90.3 | 82.3 | 97.7 | 74.7 | 86.2 | — | — | — | — | — |
| EFDM Zhang et al. (2022a) | 90.6 | 82.5 | 98.1 | 76.4 | 86.9 | — | — | — | — | — |
| $I^2$-ADRMeng et al. (2022) | 88.5 | 83.2 | 95.2 | 85.8 | 88.2 | 70.3 | 55.1 | 80.7 | 79.2 | 71.4 |
| DomainDrop Guo et al. (2023) | 89.8 | 84.2 | 98.0 | 86.0 | 89.5 | 67.3 | 60.4 | 79.1 | 80.2 | 71.8 |
| DCG Lv et al. (2023) | 90.2 | 85.1 | 97.8 | 86.3 | 89.8 | — | — | — | — | — |
| **Ours** | 90.5 | 85.2 | 98.0 | 86.8 | **90.1** | 70.6 | 59.1 | 79.3 | 82.4 | **72.9** |

## 4.2 CROSS-DOMAIN GENERALIZATION

**Datasets**: To verify the effectiveness of our method on tackling unknown data distribution shifts, we evaluate the classification performance on two widely-used cross-domain generalization benchmarks, PACS Li et al. (2017a) and Office-Home Venkateswara et al. (2017). For PACS, it contains four different style domains (*Art Painting*, *Cartoon*, *Photo*, and *Sketch*) and seven common classes with a total number of 9991 images. For Office-Home, it contains 15500 images of 65 classes collected from 4 different domains (*Art*, *Clipart*, *Product*, and *Real*). Following the protocols in Zhou et al. (2021), we use the same *leave-one-domain-out* protocol (train on three domains and test on the rest one) and ResNet-18/50 as the backbones for all methods. Several state-of-the-art domain generalization methods (listed in Table 4.2) are compared, and we show their results reported in the corresponding paper.

**Comparison Results with State-of-the-art Methods**: As shown in Table 4.2, for both datasets, our method outperforms the other cross-domain classification approaches by improving the overall improvement under different backbone networks. These results demonstrate the utilization of phantom uncertainty modeling can better handle unknown domain shifts by obtaining more discriminative and robust domain-invariant features.

## 5 CONCLUSION

The utilization of variational inference in uncertainty modeling often results in a propensity to underestimate the distribution's variance. Consequently, this tendency leads to overconfident yet incorrect predictions, particularly when dealing with noisy or ambiguous data. By introducing a novel approach focused on modeling uncertainty of artificially constructed "phantoms," which harness discriminative information gleaned from class-contextual samples, we effectively tackle the issue of underestimating uncertainty. We demonstrate the advantage of our method by showcasing the promising robustness and generalization capabilities in challenging domains like noisy label learning and cross-domain generalization, underscoring its potential to contribute to the development of more reliable and trustworthy machine learning models.

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

## A   APPENDIX A: PROVE OF EQ. 6

$$
\begin{aligned}
\log p&(y, y^+, y^- | \boldsymbol{x}, \boldsymbol{x}^+, \boldsymbol{x}^-) \\
&= \log \int p(y, y^+, y^-, \hat{\boldsymbol{z}} | \boldsymbol{x}, \boldsymbol{x}^+, \boldsymbol{x}^-) d\hat{\boldsymbol{z}} \\
&= \log \int q(\hat{\boldsymbol{z}} | \boldsymbol{x}, \boldsymbol{x}^+, \boldsymbol{x}^-) \frac{p(y, y^+, y^-, \hat{\boldsymbol{z}} | \boldsymbol{x}, \boldsymbol{x}^+, \boldsymbol{x}^-)}{q(\hat{\boldsymbol{z}} | \boldsymbol{x}, \boldsymbol{x}^+, \boldsymbol{x}^-)} d\hat{\boldsymbol{z}} \\
&\geq \int q(\hat{\boldsymbol{z}} | \boldsymbol{x}, \boldsymbol{x}^+, \boldsymbol{x}^-) \log \frac{p(y, y^+, y^-, \hat{\boldsymbol{z}} | \boldsymbol{x}, \boldsymbol{x}^+, \boldsymbol{x}^-)}{q(\hat{\boldsymbol{z}} | \boldsymbol{x}, \boldsymbol{x}^+, \boldsymbol{x}^-)} d\hat{\boldsymbol{z}} \qquad \text{(Jensens' inequality)} \\
&= \int q(\hat{\boldsymbol{z}} | \boldsymbol{x}, \boldsymbol{x}^+, \boldsymbol{x}^-) \log \frac{p(y, y^+, y^- | \hat{\boldsymbol{z}}) p(\hat{\boldsymbol{z}} | \boldsymbol{x}, \boldsymbol{x}^+, \boldsymbol{x}^-)}{q(\hat{\boldsymbol{z}} | \boldsymbol{x}, \boldsymbol{x}^+, \boldsymbol{x}^-)} d\hat{\boldsymbol{z}} \\
&= \int q(\hat{\boldsymbol{z}} | \boldsymbol{x}, \boldsymbol{x}^+, \boldsymbol{x}^-) \log p(y, y^+, y^- | \hat{\boldsymbol{z}}) d\hat{\boldsymbol{z}} + \int q(\hat{\boldsymbol{z}} | \boldsymbol{x}, \boldsymbol{x}^+, \boldsymbol{x}^-) \log \frac{p(\hat{\boldsymbol{z}} | \boldsymbol{x}, \boldsymbol{x}^+, \boldsymbol{x}^-)}{q(\hat{\boldsymbol{z}} | \boldsymbol{x}, \boldsymbol{x}^+, \boldsymbol{x}^-)} d\hat{\boldsymbol{z}}.
\end{aligned}
$$
(17)

## B   APPENDIX B: PROVE OF EQ. 11

$$
\begin{aligned}
&\int q(\hat{\boldsymbol{z}} | \boldsymbol{x}, \boldsymbol{x}^+, \boldsymbol{x}^-) \log \frac{p(\hat{\boldsymbol{z}} | \boldsymbol{x}, \boldsymbol{x}^+, \boldsymbol{x}^-)}{q(\hat{\boldsymbol{z}} | \boldsymbol{x}, \boldsymbol{x}^+, \boldsymbol{x}^-)} d\hat{\boldsymbol{z}} \\
&= \int q(\hat{\boldsymbol{z}} | \boldsymbol{x}, \boldsymbol{x}^+, \boldsymbol{x}^-) \log \frac{p(\hat{\boldsymbol{z}} | \boldsymbol{x}, \boldsymbol{x}^+, \boldsymbol{x}^-)}{q(\hat{\boldsymbol{z}} | \boldsymbol{x}, \boldsymbol{x}^+, \boldsymbol{x}^-)} \cdot \frac{p(\boldsymbol{z} | \boldsymbol{x}) p(\boldsymbol{z}^+ | \boldsymbol{x}^+) p(\boldsymbol{z}^- | \boldsymbol{x}^-)}{p(\boldsymbol{z} | \boldsymbol{x}) p(\boldsymbol{z}^+ | \boldsymbol{x}^+) p(\boldsymbol{z}^- | \boldsymbol{x}^-)} d\hat{\boldsymbol{z}} \\
&= \int q(\hat{\boldsymbol{z}} | \boldsymbol{x}, \boldsymbol{x}^+, \boldsymbol{x}^-) \log \frac{p(\hat{\boldsymbol{z}} | \boldsymbol{x}, \boldsymbol{x}^+, \boldsymbol{x}^-)}{p(\boldsymbol{z} | \boldsymbol{x}) p(\boldsymbol{z}^+ | \boldsymbol{x}^+) p(\boldsymbol{z}^- | \boldsymbol{x}^-)} d\hat{\boldsymbol{z}} + \int q(\hat{\boldsymbol{z}} | \boldsymbol{x}, \boldsymbol{x}^+, \boldsymbol{x}^-) \log \frac{p(\boldsymbol{z} | \boldsymbol{x}) p(\boldsymbol{z}^+ | \boldsymbol{x}^+) p(\boldsymbol{z}^- | \boldsymbol{x}^-)}{q(\hat{\boldsymbol{z}} | \boldsymbol{x}, \boldsymbol{x}^+, \boldsymbol{x}^-)} d\hat{\boldsymbol{z}} \\
&= \int q(\hat{\boldsymbol{z}} | \boldsymbol{x}, \boldsymbol{x}^+, \boldsymbol{x}^-) \log \frac{p(\hat{\boldsymbol{z}} | \boldsymbol{x}, \boldsymbol{x}^+, \boldsymbol{x}^-)}{p(\boldsymbol{z} | \boldsymbol{x}) p(\boldsymbol{z}^+ | \boldsymbol{x}^+) p(\boldsymbol{z}^- | \boldsymbol{x}^-)} d\hat{\boldsymbol{z}} - \mathrm{KL}[q(\hat{\boldsymbol{z}} | \boldsymbol{x}, \boldsymbol{x}^+, \boldsymbol{x}^-) || p(\boldsymbol{z} | \boldsymbol{x})] \\
&\quad - \mathrm{KL}[q(\hat{\boldsymbol{z}} | \boldsymbol{x}, \boldsymbol{x}^+, \boldsymbol{x}^-) || p(\boldsymbol{z}^+ | \boldsymbol{x}^+)] - \mathrm{KL}[q(\hat{\boldsymbol{z}} | \boldsymbol{x}, \boldsymbol{x}^+, \boldsymbol{x}^-) || p(\boldsymbol{z}^- | \boldsymbol{x}^-)].
\end{aligned}
$$
(18)

## C   APPENDIX C: PROVE OF EQ. 14

$$
\begin{aligned}
\mathcal{L}_{\mathrm{TM}} &= -\log \frac{\exp\{\boldsymbol{z} \cdot \boldsymbol{z}^+ / \tau\}}{\exp\{\boldsymbol{z} \cdot \boldsymbol{z}^+ / \tau\} + \exp\{\boldsymbol{z} \cdot \boldsymbol{z}^- / \tau\}} \\
&= \log\{1 + \exp((\boldsymbol{z} \cdot \boldsymbol{z}^+ - \boldsymbol{z} \cdot \boldsymbol{z}^+)/\tau)\} \\
&\approx \exp\{(\boldsymbol{z} \cdot \boldsymbol{z}^- - \boldsymbol{z} \cdot \boldsymbol{z}^+)/\tau\} \qquad\qquad \text{(Taylor expansion of log)} \\
&\approx 1 + \frac{1}{\tau} \cdot (\boldsymbol{z} \cdot \boldsymbol{z}^- - \boldsymbol{z} \cdot \boldsymbol{z}^+) \qquad\qquad \text{(Taylor expansion of exp)} \\
&= 1 - \frac{1}{2\tau} \cdot \left(\|\boldsymbol{z} - \boldsymbol{z}^-\|^2 - \|\boldsymbol{z} - \boldsymbol{z}^+\|^2\right) \qquad \text{(Normalized embeddings)} \\
&\propto \|\boldsymbol{z} - \boldsymbol{z}^+\|^2 - \|\boldsymbol{z} - \boldsymbol{z}^-\|^2 + 2\tau
\end{aligned}
$$
(19)

## D   APPENDIX: MORE EXPERIMENTAL RESULTS

Here, we conduct more ablation analysis to investigate our method further.

**The Visualization of Learned Uncertainty Values**: We conduct the following visualization experiments to verify that our method can better quantify the uncertainty of these error-prone samples. Fig. 3 compares the uncertainty score predictions and recognition results of our method and the baseline variational-based method DUL Chang et al. (2020) for the same case. From top to bottom of the blank in the right of each image, the ground-truth label, the prediction results of our method, and that

of the baseline method DUL are shown. For some examples, although DUL predicts relatively small uncertainty values for them by considering them as easy cases, DUL still can not correctly classify them, which demonstrates the uncertainty estimation results deviate from the factual results. While our method performs well on the examples with either high uncertainty scores or low scores. This demonstrates that our method can not only accurately estimate the uncertainty of the images but also correctly classify them by eliminating the adverse influence of uncertainty.

**The t-SNE Visualization of Learned Features**: To further verify our method is robust to data noise, we utilize t-SNE Van der Maaten & Hinton (2008) to visualize the learned features of our method under different noise levels on RAF-DB. As shown in Fig. 4, the feature embedding space learned by our method demonstrates the intra-category compactness and inter-category separability even if the noise level is at a high level of 30%. This visualization result is consistent with previous quantitative results, demonstrating that the proposed method is robust to data noise.

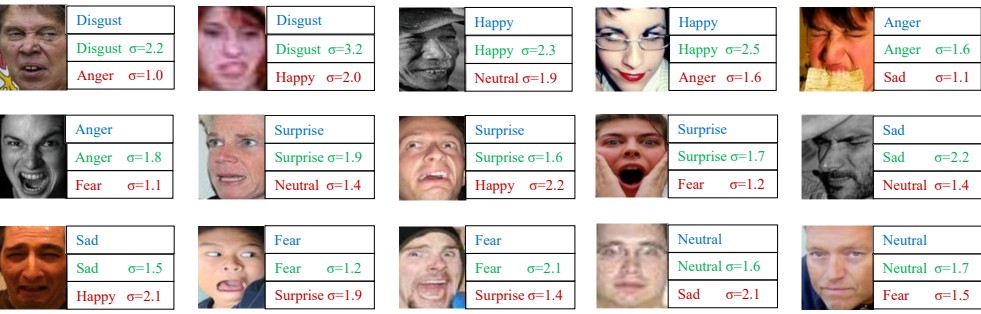

Figure 3: The visualization results of our method and the baseline method DUL on hard samples in RAF-DB.

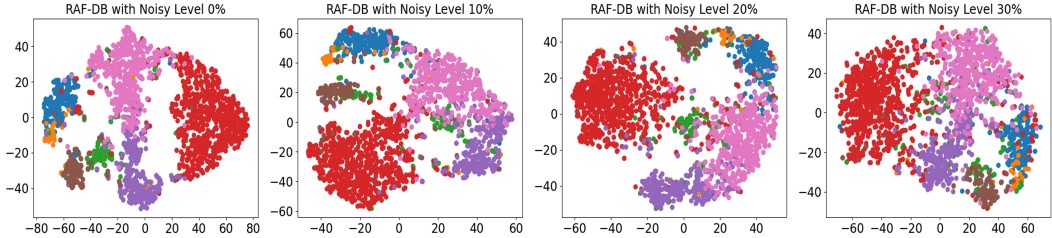

Figure 4: The t-SNE results of test feature embedding learned by our method on RAF-DB with the noise level of 0-30%.

**Computation overhead**: We compare the computation overhead with our baseline DUL Chang et al. (2020), under the same network architecture. On the RAF-DB dataset, the average training time for each epoch is 17.28s for DUL and 18.24s for our method. The extra overhead is marginal, given the consistent performance improvement in various experiments.

