# OpenReview forum: "Class-Context-Aware Phantom Uncertainty Modeling"
_ICLR.cc/2024/Conference — Submitted to ICLR 2024_

### Official Review · Reviewer_QLP6 · 2023-10-30

**Soundness:** 2 fair
**Presentation:** 2 fair
**Contribution:** 3 good
**Rating:** 5
**Confidence:** 3

**Summary:**

This study proposed the improved Aleatoric uncertainty evaluation by focusing on ambiguous class labels to distinguish the correctly specified labels. The poposed method is implemented by the famwrok of variational inference and improves existing mean-field  Gaussian-based variational inference.

**Strengths:**

The paper adeptly merges the settings of variational inference and the inherent settings of class labels. Particularly, the approach of combining three types of label information in a Bayesian manner, as shown in Eq 7, seems novel and very intriguing. Furthermore, the implementation is noteworthy for its compatibility and integration with existing methods, showcasing a high degree of extensibility.

**Weaknesses:**

There are unclear points in the derivation of the equations, as mentioned in the question. Moreover, as the variational inference, the proposed method involves numerous approximations rather than relying on the Evidence Lower Bound (ELBO), it is not trivial that the proposed objective function is "proper" or "appropriate" objective function, yet this issue is not discussed in the paper.

Another concern is about creating phantoms in Equation 5 using a mini-batch. While mini-batching is computationally efficient, I am not sure whether it provides sufficient boundary information about ambiguous class labels within a selected minibatch. This raises doubts about the method's effectiveness being heavily dependent on mini-batch selection.

The paper also mentions using self-paced learning in experiments, yet it lacks a thorough discussion on its critical necessity for the proposed method. This aspect needs further exploration and clarification to strengthen the paper’s validity.

**Questions:**

In addition to the weakness,
- How Eq 12 is derived?

---

### Official Review · Reviewer_najj · 2023-10-30

**Soundness:** 2 fair
**Presentation:** 2 fair
**Contribution:** 2 fair
**Rating:** 3
**Confidence:** 4

**Summary:**

The authors address the challenge of underestimating variance in uncertainty quantification by introducing "phantom objects".
These are not real objects from the dataset, but their variance serves as an upper bound to the variance of the posterior of the object in focus.

The authors demonstrate that the proposed approach helps rectify the issue of uncertainty underestimation and show that the proposed method has superior robustness compared to considered baselines.

**Strengths:**

1. The paper is clear and easy to follow, except for some misprints/errors.
2. The selection of baselines considered is extensive.

**Weaknesses:**

I find the text itself pretty raw, and I think a revision should be done.

- There are misprints:
1. In Eq. 1: The representation g: f(x) -> Y where Y is a set of labels {1,..., K} suggests that g should extract the argmax from the softmax output. However, in Eq. (1), g is depicted as the last FC layer. Thus, g should represent a mapping from R^d -> R^K. Otherwise, under Softmax, you have class labels, not logits.
2. In Eq. 9: It seems $\hat{\sigma}^2$ should be changed to $\hat{\sigma}^{-2}$, as in Eq. (8).

- Not accurate statements:
1) Page 3: "Epistemic uncertainty ... is incurred by a lack of knowledge, usually due to insufficient training data". It is partially true, but also epistemic uncertainty is related to model misspecification (see for example [1]).
2) Page 3: On the other hand, our focus is more on aleatoric uncertainty ... which mainly comes from ... OOD outliers and unknown domain shifts". Can you please elaborate on what you mean by saying OOD outliers influence aleatoric uncertainty?

- Method
1) The method itself is based on the intuition that embeddings are well separated and there is no so-called "feature collapse" (see for example  [2, 3, 4]). But this might not hold in practice. Thus, some special treatments, like spectral normalization or gradient penalty, may help. In any case, the discussion is required
2) It is not mentioned what norm was used to compute eq. 5. In any case, the geometry might be too complicated, and standard p-norms could be suboptimal.
3) On page 5: "2) the negative class context $x^{-}$, visually similar to $x$". Again, if there is a collapse of features, these intuitions that "close in image space -> close in embedding space" do not hold anymore.
4) I think there is a missing opportunity to visualize these $x^{-}$ and $x^{+}$ samples.

- Experiments:
1) It is not clear what Table 3 (which is referred to as Table 4.2 in the text) represents. Is it Accuracy? ROC-AUC?
2) Page 9: "... improving the overall improvement"
3) I think it is worth discussing baselines more in detail. At least in the appendix.
4) The section in the Appendix with computation overhead seems to be never referenced.

---

[1] Lahlou, Salem, et al. "Deup: Direct epistemic uncertainty prediction." arXiv preprint arXiv:2102.08501 (2021).

[2] Van Amersfoort, Joost, et al. "Uncertainty estimation using a single deep deterministic neural network." International conference on machine learning. PMLR, 2020.

[3] Mukhoti, Jishnu, et al. "Deep deterministic uncertainty for semantic segmentation." arXiv preprint arXiv:2111.00079 (2021).

[4] Kotelevskii, Nikita, et al. "Nonparametric Uncertainty Quantification for Single Deterministic Neural Network." Advances in Neural Information Processing Systems 35 (2022): 36308-36323.

**Questions:**

Can you elaborate on the derivation of equation 12?

---

### Official Review · Reviewer_QjyL · 2023-10-31

**Soundness:** 2 fair
**Presentation:** 3 good
**Contribution:** 2 fair
**Rating:** 3
**Confidence:** 4

**Summary:**

Authors introduce a novel approach to incorporate uncertainty modeling within neural networks using variational inference. Instead of directly characterizing the probability distribution of input representations, they present a unique strategy that captures the probability distribution within a latent space referred to as the "class-context aware phantom space." In this method, the distribution of latent variables is learned by leveraging variations within different classes and samples that share similarities but possess contrasting labels.

Furthermore, the authors introduce a specialized loss function to the train the distribution in phantom space, which acts as a lower bound for the conventional variational inference loss. Finally, the authors provide empirical evidence of the effectiveness of their uncertainty estimation approach through a series of experiments.

**Strengths:**

Variational inference is a widely employed technique for estimating uncertainty, but it is without limitations. This work seeks to tackle a significant limitation in variational inference methods.

Authors present a novel idea of learning distribution in the latent space, which incorporates contextual information. This was done by  leverages both intra-class variability and the diversity of labels for similar data points.

To facilitate the training of this distribution, the authors derive an Evidence Lower Bound (ELBO) loss function, and they empirically validate the effectiveness of their proposed method.

**Weaknesses:**

I believe there are some shortcomings in the paper's experimental approach. Firstly, it's somewhat unclear why aleatoric uncertainty would be beneficial in detecting and filtering out Out-of-Distribution (OOD) data. Secondly, the proposed method doesn't seem to exhibit substantial improvements when compared with existing methods. Moreover, the absence of confidence intervals for the results is notable, and this is crucial, especially considering the closely matched performances of various methods.


In the concluding paragraph of Section 2, the authors point out that a key issue associated with Variational Inference (VI) is the potential for VI to result in posterior collapse and subsequently lead to an underestimation of variance. They assert that their proposed method does not suffer from this limitation.  In the same paragraph, authors state that "variance linked to these phantoms establishes a minimum threshold for estimating the actual variance of interest." This statement appears to carry a potential contradiction. We kindly request the authors to provide additional clarification to reconcile this seeming discrepancy.

It is not clear if and why modeling distribution in phantom space can tackle the variance underestimation problem of VI. It would be useful if authors can provide some insights regrading this.

Based on Equation 5, it looks like the proposed method is highly susceptible to label noise, as x^+ and x^- are highly susceptible to noise. In most practical settings, label noise might be unavoidable. Would it be better if the authors actually used percentile methods (ex: datapoint which is say 99 percentile away) to make this more robust.

Above Eq 7, an independence assumption about z, z^+, and z^- was made. It would be better to make this assumption more explicit. More importantly, it is not clear why z, z^+, z^- being conditionally independent of each other given x, x^+, x^- can lead to q(z|x, x^+, x^-) = q(z|x). Can authors explain this in more detail?

**Questions:**

In addition to the points mentioned in the weakness section, my comments are:

In the second paragraph of Section 1, the authors underscore the rationale behind their research, stating that the modeling of aleatoric uncertainty has the potential to rectify overly confident predictions when dealing with unfamiliar inputs (or  OOD data). However, it remains unclear how aleatoric uncertainty can be instrumental in addressing OOD data predictions. We kindly request the authors to clarify this.

Authors present a novel method which makes use of positive and negative class content samples to construct distribution of "phantom" variables. Can authors kindly provide an intuition for why this doesn't happen with the default VI method already?

In the paragraph below equation 11, phrase "we introduce a valid assumption" was used, but the assumption was not discussed. Can authors kindly look into this?

In the experimental section, accuracy is used as a metric.However, it's worth noting that accuracy is often considered a high Signal-to-Noise Ratio (SNR) metric, meaning it provides more informative results when the dataset has minimal noise. It's currently unclear whether the datasets under consideration can be categorized as high SNR datasets, and whether accuracy is a suitable indicator for determining the adequacy of uncertainty modeling. We kindly request the authors to provide their insights on this matter for better context and interpretation of the results.

---

### Official Review · Reviewer_R4BK · 2023-10-31

**Soundness:** 3 good
**Presentation:** 3 good
**Contribution:** 3 good
**Rating:** 5
**Confidence:** 3

**Summary:**

This paper proposes a new type of uncertainty-aware classification model. The method augments the classification model with a continuous latent variable, which introduces additional uncertainty into the mapping between the input and the label. The paper then argues that, using mean-field VI for inferring the representation could potentially lead to under-estimation of the variance, resulting in overconfidence. Therefore the paper further proposes to incorporate the *inter-class variance among samples in a batch* into the representations, the resulting representation, *phantom*, is an average of the representations from the most and least ambiguous and the test sample. In such a way, the resulting classifier becomes more robust against label noise and domain shift. The paper shows experiment results in noisy label learning and domain generalization experiments, in both settings, the proposed method shows improvement compared with existing approaches.

**Strengths:**

- The method is technically sound and the experiments are thorough.

- The method is supported by rigorous derivation and math principles.

- The experiment settings are clearly stated, which helps reproducibility.

**Weaknesses:**

- The paper is motivated by "mean-field VI underestimates the variance". The motivation is sensible however the paper did not provide any evidence verifying this motivation.

- The method does show performance improvement but I am worried that the performance could be sensitive to the setting of the three hyper-parameters. It is also not clear how one should set these hyper-parameters.

- The paper does not mention the "complexity" of the proposed method, e.g. does the method require extra training time, hyper-parameter tuning effort, memory cost, etc. compared with baseline approaches?

- It would be good if the paper could have more uncertainty-related results, the current version does not have any direct results demonstrating the uncertainty acquired through the method. A simple example would be to show the decision boundary on two moons dataset or XOR dataset.

**Questions:**

- How many samples of z does the experiment use to construct predictions? Since the method constructs a distribution of z, then one can sample multiple z and construct an ensemble for prediction.

- I imagine incorporating uncertainty into the classifier could enhance the calibration and out-of-distribution detection ability, but why would uncertainty help cross-domain generalization?

- The paper mentions "Best Linear Unbiased Estimation (BLUE) Henderson (1975)" several times but I didn't understand the point, is it related to the theoretical guarantee of the proposed method?

---

### Meta-Review · Area_Chair_AzY9 · 2023-12-05

**Metareview:**

The paper focuses on mitigating the issue of underestimating the uncertainty associated with the input data by inferring the distribution of their respective phantoms, which are derived by leveraging class-contextual information. Experiments have been conducted to assess the robustness and generalization capabilities of the proposed method in noisy label learning and cross-domain generalization settings.

Reviewers have expressed concerns about the missing of concrete evidences and/or deeper insights on key aspects of the proposed approach, making it less convincing. In addition, some of the assumptions may not hold true in practice and the derivation of important equations are not clearly presented. Reviewers are also concerned that the proposed method does not show clear improvement over existing baselines or the improvement could be sensitive to the setting of the hyper-parameters.

Since the authors did not provide any rebuttal, the above major concerns remain to be addressed. The authors are suggested to carefully consider the constructive comments from individual reviewers to further improve their paper.

**Justification For Why Not Higher Score:**

Since the authors did not provide a rebuttal, the major concerns from the reviewers were not addressed.

**Justification For Why Not Lower Score:**

N/A

---

### Decision · Program_Chairs · 2024-01-16

Reject